# Long-term outcomes following drug-coated balloons versus thin-strut drug-eluting stents for treatment of in-stent restenosis in Chronic Kidney Disease (CKD Dragon-Registry)

Rafał Januszek[1]*, Marta Chamera[2], Sylwia Iwańczyk[3,4], Fabrizio D'Ascenzo[5], Łukasz Kuźma[6], Brunon Tomasiewicz[7], Piotr Niezgoda[8], Rafał Wolny[9], Mariusz Kowalewski[10], Maciej Wybraniec[11], Krzysztof Reczuch[7], Sławomir Dobrzycki[6], Ovidio De Filippo[5], Artur Pawlik[12], Karol Kasprzycki[12], Kamil Skowron[12], Stanisław Bartuś[12], Maciej Lesiak[4], Mariusz Gąsior[13], Jacek Kubica[8], Tomasz Pawłowski[14], Robert J. Gil[14], Piotr Waciński[15], Francesco Bruno[5], Bernardo Cortese[3,16], Wojciech Wojakowski[2], Wojciech Wańha[2,3,11]

1 Faculty of Medicine and Health Sciences, Andrzej Frycz Modrzewski Cracow University, Kraków, Poland, 2 Department of Cardiology and Structural Heart Diseases, Medical University of Silesia, Katowice, Poland, 3 DCB Academy, Milano, Italy, 4 1st Department of Cardiology, Poznan University of Medical Sciences, Poznan, Poland, 5 Division of Cardiology, Department of Internal Medicine, Città della Salute e della Scienza, University of Turin, Turin, Italy, 6 Department of Invasive Cardiology, Medical University of Bialystok, Bialystok, Poland, 7 Centre for Heart Disease, University Hospital Wroclaw Department of Heart Disease, Wroclaw Medical University, Wrocław, Poland, 8 Department of Cardiology and Internal Medicine, Collegium Medicum, Nicolaus Copernicus University, Bydgoszcz, Poland, 9 Department of Interventional Cardiology and Angiology, National Institute of Cardiology, Warsaw, Poland, 10 Clinical Department of Cardiac Surgery and Transplantology, National Medical Institute of the Ministry of Interior and Administration, Warsaw, Poland, 11 1st Department of Cardiology, School of Medicine in Katowice, Medical University of Silesia, Katowice, Poland, 12 Jagiellonian University Medical College, Kraków, Poland, 13 Third Department of Cardiology, Medical University of Silesia, Zabrze, Poland, 14 Department of Cardiology, National Medical Institute of the Ministry of Interior and Administration, Warsaw, Poland, 15 Clinical Department of Interventional Cardiology, Medical University of Lublin, Lublin, Poland, 16 Harrington Heart & Vascular Institute, University Hospitals Cleveland Medical Center, Cleveland, Ohio, United States of America

* jaanraf@interia.pl

## Abstract

We sought to investigated the outcomes of patients with chronic kidney disease (CKD) and drug-eluting stent (DES)-in-stent restenosis (ISR) undergoing percutaneous coronary intervention (PCI) with a drug-coated balloon (DCB) or thin strut drug-eluting stent (thin-DES). Consecutive patients with DES-ISR who underwent PCI with a thin-DES or a paclitaxel-coated DCB for DES-ISR were enrolled. The primary outcome was target lesion revascularization (TLR), while the secondary was target vessel revascularization (TVR) and device-oriented composite endpoint (DOCE). The pooled analysis included 1,317 patients, with 585 (44.42%) treated using a thin-DES and 732 (55.58%) by DCB. In the crude analysis of CKD patients (n = 286) undergoing PCI for ISR, thin-DES vs. DCB showed similar outcomes for TLR (hazard ratio [HR] = 0.94, 95% confidence interval [CI] = 0.44–2.00; p = 0.873),

**Data availability statement:** All relevant data are within the manuscript and its Supporting Information files.

**Funding:** The author(s) received no specific funding for this work.

**Competing interests:** The authors have declared that no competing interests exist.

TVR (HR = 0.82, 95% CI = 0.44–1.55; p = 0.542), MI (HR = 0.71, 95% CI = 0.34–1.46; p = 0.348) and DOCE (HR = 0.71, 95% CI = 0.36–1.40; p = 0.325). After propensity score matching (n = 184), the HRs remained non-significant for TLR (0.52, 95% CI = 0.21–1.29; p = 0.159), TVR (0.54, 95% CI = 0.24–1.01; p = 0.134), MI (0.56, 95% CI = 0.24–1.32; p = 0.183), TV-MI (0.56, 95% CI = 0.09–3.39; p = 0.528), cardiac death (0.63, 95% CI = 0.10–3.81; p = 0.615), and DOCE (0.45, 95% CI = 0.19–1.04; p = 0.062). In conclusion, in CKD patients undergoing PCI for ISR, thin-DES treatment was associated with a numerical reduction in TLR, TVR, and DOCE compared with DCB. However, these differences did not achieve statistical significance in the crude or propensity score-matched analyses.

## Introduction

One of the main disadvantages of percutaneous coronary revascularization with stenting is the risk of in-stent restenosis (ISR) during the follow-up period [1]. A ≥ 50% reduction in luminal diameter within the stented segment defines binary angiographic ISR [2]. The time from percutaneous coronary intervention (PCI) to ISR occurrence is determined by several factors, including the type of PCI, implanted stent [3], antiplatelet drug used and treatment duration [4], target lesion, plaque-stabilizing treatment, and the control and number of concomitant modifiable atherosclerosis risk factors [5], as well as non-modifiable elements such as congenital and environmental factors [3–5]. Despite meaningful improvements in DES technologies and modifiable risk factor control, ISR occurs at a rate of 1−2% per year with contemporary DES platforms [6]. Such lesions are often seen in clinical practice and, compared to *de novo* lesions, are associated with a higher incidence of adverse cardiac events regardless of the interventional approach [1,7,8]. Registry data shows that PCI for ISR is performed in around 10% of all PCI cases [9,10], with DES and drug-coated balloon (DCB) use as the only methods currently recommended for ISR treatment [11,12]. The availability of data comparing the outcomes of ISR treatment in patients with renal failure is very limited, and there are no clear recommendations as to which treatment method is better in this subgroup of patients. A study by Mahfound et al. [13] found that when using PCI for small coronary artery disease, the primary outcomes were similar over three years in DCB and DES patients (hazard ratio [HR] = 0.98, 95% confidence interval [CI] 0.67–1.44; p = 0.937), and patients with and without CKD (HR = 1.18; 95% CI = 0.76–1.83; p = 0.462). Rates of cardiac and all-cause death were significantly higher among CKD patients but were not affected by DCB or DES treatment [13]. However, major bleeding events were lower for DCB than DES (12 vs. 3, HR = 0.26, 95% CI = 0.07–0.92; p = 0.037) and not influenced by the presence of CKD [13]. Several randomized trials and retrospective studies have shown comparable long-term outcomes between DES and DCB in non-selected ISR patients [14], though less is known about the impact of CKD on DCB and DES treatment in ISR patients. Lee et al. found that CKD patients with ISR undergoing DCB angioplasty had a significantly higher risk of adverse events compared to patients

with preserved renal function. In contrast, subgroups with mild to moderate CKD did not display this difference [15]. The study concluded that different revascularization strategies may be considered for ISR patients with severe CKD or end-stage renal disease (ESRD) [15].

Therefore, in the [resented study we assessed the long-term safety and efficacy of DCB use versus a thin strut DES (thin-DES) in CKD patients with ISR.

## Materials and methods

### Study design, objectives, and patient selection

This observational, multicenter study based on the CKD DRAGON Registry enrolled all consecutive patients who underwent PCI with a new-generation DES or DCB for coronary DES-ISR between February 2008 and October 2021. Subsequently, patients within the DES and DCB groups were assigned to one of two subgroups depending on an estimated glomerular filtration rate (eGFR) < 60 mL/min/1.73m². Exclusion criteria included concomitant DES and DCB treatment, saphenous vein graft ISR, cardiogenic shock, thrombolysis before PCI, suboptimal or failed PCI for target lesions, and 12-month follow-up not available. Fig 1 depicts the patient flow chart.

Coronary angiography assessed ISR, defined as a ≥ 50% reduction in luminal diameter within the previously stented segment or the vessel segments 5 mm proximal and distal to the stent. In case of doubts regarding the significance of stenoses in the angiographic examination, confirmation or exclusion relied on a physiological assessment. The Mehran classification defined ISR patterns [16].

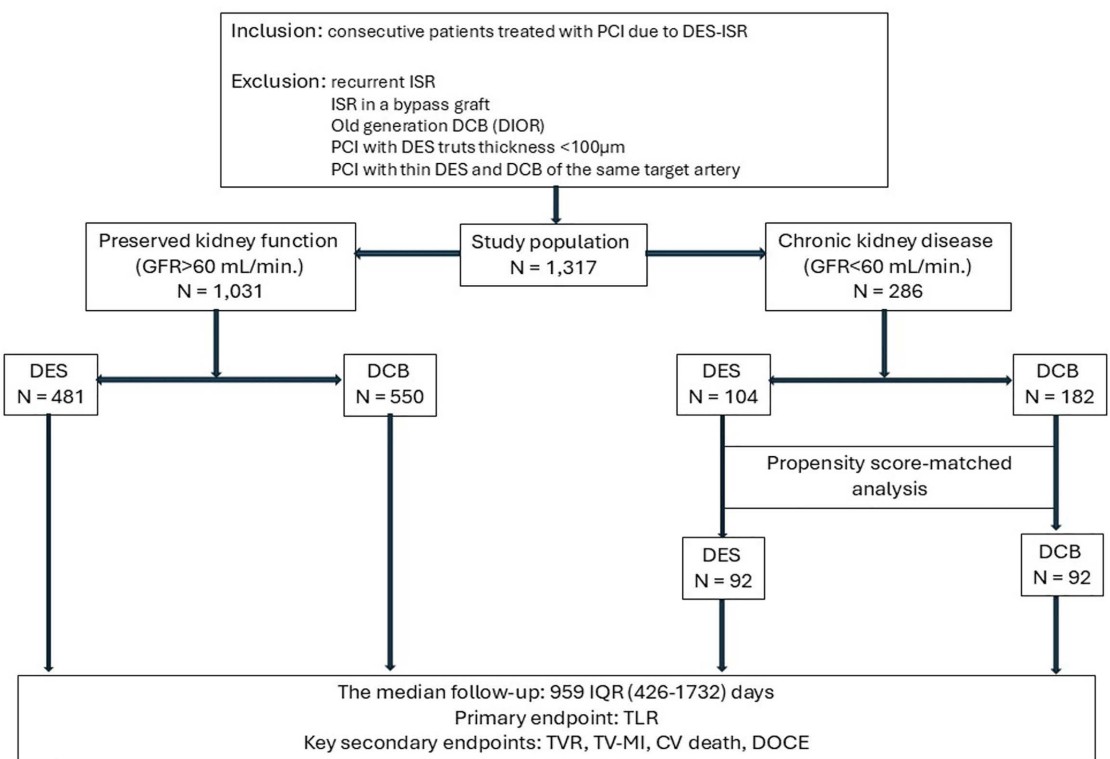

**Fig 1. Patient flow chart.**

## Procedure description

All patients were preloaded with a P2Y12-antagonist and aspirin before coronary angioplasty. Unfractionated heparin was given according to the standard hospital practice. Additional antithrombotic regimens, including glycoprotein IIb/IIIa inhibitors, were used at the discretion of the treating physicians. Interventions were performed with 6 or 7 French guiding catheters, using the radial approach in more than 90% of patients. Intravascular ultrasound (IVUS) or optical coherence tomography (OCT) was optional but recommended to assess the ISR mechanism. Pre-dilatation of the target lesion was mandatory, using an uncoated balloon catheter with a diameter 0.5 mm smaller than, or similar to the size of, the reference vessel diameter or the diameter of the previously implanted restenotic stent. The operator used atherectomy methods and specially modified balloons for severe calcifications, such as scoring or cutting balloons. The recommended study DCB inflation time was 30–60 seconds at nominal pressure. Operator and center experience determined DCB choice.

The procedure was successful if final a flow grade 3 thrombolysis in myocardial infarction (TIMI) without flow-limiting dissection was obtained, final diameter stenosis was below 30% by visual estimation, and no in-hospital major adverse cardiovascular events (MACE) occurred. Dual antiplatelet therapy (DAPT) was generally continued orally for at least six months in stable or 12 months in acute coronary syndrome (ACS) patients, followed by aspirin or clopidogrel alone.

## Inclusion criteria

The CCS DRAGON Registry is a multicenter initiative involving consecutive patients with DES-ISR and CKD manifestation treated with a paclitaxel-DCB or a thin-DES between February 2008 and October 2021. Exclusion criteria were DCB and thin-DES use during the same procedure, PCI of a bypass graft, and recurrent ISR. Thin strut stents were defined as those with strut thickness <100 μm. The following DES were used: Alex (Balton, Warsaw, Poland), Orsiro (Biotronik AG, Bulach, Switzerland), Promus (Boston Scientific, MA, USA), Resolute (Medtronic CardioVascular, CA, USA), Synergy (Boston Scientific, MA, USA), Ultimaster (Terumo Corporation, Tokyo, Japan), and Xience (Abbott Vascular Devices, CA, USA). The paclitaxel-DCB used were Agent (Boston Scientific, MA, USA), Elutax (Aachen Resonance GmbH, Aachen, Germany), Essential (iVascular, Barcelona, Spain), IN.PACT (Medtronic Vascular, CA, USA), Pantera Lux (Biotronik AG, Buulach, Switzerland), Restore DEB (Cardionovum GmbH, Bonn, Germany), and SeQuent Please Neo (B.Braun Interventional Group, Ltd, Melsulgen, Germany).

Patient data were anonymized at each center, combined into one database, and statistically analyzed as one cohort. Cardiovascular risk factors, clinical presentation, and angiographic characteristics were recorded, along with the parameters of the implanted stents. The above data were derived from electronic patient records at each center. The institutional review board approved the study protocol, though, due to the retrospective nature of the study, no written informed consent was needed. Patient data was protected according to the requirements of Polish law, the General Data Protection Regulation (GDPR), and hospital standard operating procedures. The study was conducted in accordance with the Declaration of Helsinki and registered at www.ClinicalTrials.gov. The data supporting the findings of this study are available from the corresponding author upon reasonable request.

## Primary and secondary endpoints

Outcome data were obtained from clinical assessments, telephone consultations, or primary care physicians and recorded online or from the central database of the National Health Fund Service of the Ministry of Health. No patients were lost to follow-up. The primary efficacy endpoint was target lesion revascularization (TLR). The secondary endpoints were device-oriented composite endpoint (DOCE) (defined as a composite of cardiac death, TLR, and target vessel myocardial infarction [MI]), target vessel revascularization (TVR), MI, and cardiac death. TVR and TLR were defined according to the definitions of endpoints for clinical trials.

## Statistical analysis

All statistical analyses used a two-sided significance level of α = 0.05. The Shapiro-Wilk test assessed the distribution of continuous variables. Data were presented as medians (Me) and interquartile range (IQR) for non-normally distributed continuous variables and as the number of cases (n) and percentage (%) for categorical variables. For nominal variables, Pearson's chi-squared test and Fisher's exact test determined statistically significant differences between the two groups, while the Wilcoxon rank-sum test compared numerical variables between two independent groups. Kaplan-Meier curves were used for graphical presentations of time-dependent variables, and a log-rank test was performed for between-group comparisons. Propensity score-matching (PSM) was conducted using the nearest neighbor algorithm to ensure comparability between DES and DCB cohorts. The matching process accounted for a comprehensive set of core baseline variables, including sex, age, atrial fibrillation, arterial hypertension, diabetes mellitus, smoking status, previous MI, prior coronary artery bypass grafting (CABG), peripheral artery disease, left ventricular ejection fraction (LVEF), degree of stenosis, and lesion characteristics (bifurcation and calcification) (S1 Fig). Cox regression analysis evaluated long-term follow-up event rates for TLR, target lesion vascularization (TLV), DOCE, MI, target vessel MI, and all-cause mortality in both the unmatched population and the PSM CKD group.

The proportional hazards assumption for the Cox models was verified using the Schoenfeld residuals test (cox.zph function in the R survival package, version 3.7.0). For all primary and secondary endpoints in both the unmatched and propensity score-matched analyses, the test produced p-values exceeding significance level (α = 0.05), confirming no significant departures from proportionality.

This analysis was exploratory and hypothesis-generating in nature, with no adjustments applied for multiple comparisons across the primary and secondary endpoints to preserve sensitivity in detecting potential signals, albeit at the risk of increased Type I error.

Analyses employed the R Statistical language (version 4.3.3), using the rio (version 1.2.1), cobalt (version 4.5.5), MatchIt (version 4.5.5), sjPlot (version 2.8.15), parameters (version 0.22.2), performance (version 0.12.3), report (version 0.5.8), ggsurvfit (version 1.1.0), gtsummary (version 1.7.2), survival (version 3.7.0), MASS (version 7.3.60.0.1), ggplot2 (version 3.5.0), and dplyr (version 1.1.4) packages [17–31].

## Results

The non-matched cohort comprised 1317 patients, and before PSM, there were 1031 patients with preserved kidney function (481 treated with DES and 550 with DCB), while 286 patients had impaired function (104 patients treated with DES and 182 with DCB). After PSM, 184 patients remained in the CKD group, and both groups consisted of 92 patients. Before PSM, there were 14 patients in the DCB group undergoing dialysotherapy and nine in the DES group. After PSM, six patients remained in the DCB group, and all nine remained in the DCB group. The median follow-up was 959 days (IQR = 426–1732) (Fig 1).

### Patient characteristics

Before PSM, the CKD patients treated with DES were significantly older (p = 0.019), comprised fewer males (p = 0.025), and suffered less from diabetes mellitus (p = 0.007) and arterial hypertension (p = 0.016) than the DCB-treated group. However, such differences were not found in those with preserved kidney function (Table 1). Meanwhile, patients with preserved kidney function treated with DES had a lower history of CABG (p = 0.002) and peripheral artery disease (p < 0.001) than the DCB group but experienced ST-elevation MI (STEMI) (p = 0.004) and atrial fibrillation (p = 0.002) more often and had a lower LVEF (p < 0.001), unlike patients with impaired kidney function, where these differences where not significant (Table 1). After PSM, patients in the CDK group treated with DES presented with a history of coronary artery disease (CAD) more often (p = 0.001), while other indices did not differ significantly between subgroups (Table 1).

**Table 1. Patient characteristics, risk factors, and clinical presentation according to device type.**

| | Non-matched cohort, n=1,317 | | | | | | Propensity score-matched groups, n=184 | | |
| | Preserved kidney function, n=1,031 | | | CKD, n=286 | | | CKD, n=184 | | |
| | Thin-DES n=481 (46.65%) | DCB n=550 (53.35%) | p-value | Thin-DES n=104 (36.36%) | DCB n=182 (63.74%) | p-value | Thin-DES n=92 (50.00%) | DCB n=92 (50.00%) | p-value |
|---|---|---|---|---|---|---|---|---|---|
| **Demographic data** | | | | | | | | | |
| Age, years, Me (IQR) | 65.00 (58.36, 72.00) | 67.00 (60.00, 72.00) | 0.120[A] | 75.00 (68.75, 81.17) | 72.00 (65.00, 78.00) | **0.019**[A] | 74.61 (68.00, 79.25) | 74.00 (65.35, 81.00) | 0.965[A] |
| Male, n (%) | 340 (70.69%) | 401 (72.91%) | 0.428[B] | 57 (54.81%) | 124 (68.13%) | **0.025**[B] | 54 (58.70%) | 60 (65.22%) | 0.362[B] |
| BMI, Me (IQR) | 28.11 (25.96, 31.10) | 28.73 (25.95, 31.63) | 0.408[A] | 28.23 (25.86, 31.20) | 27.58 (25.21, 30.50) | 0.496[A] | 28.23 (25.73, 31.53) | 28.06 (25.50, 31.06) | 0.711[A] |
| **CAD history** | | | | | | | | | |
| Previous MI, n (%) | 307 (63.83%) | 338 (61.45%) | 0.433[B] | 60 (57.69) | 118 (64.84%) | 0.231[B] | 57 (61.96%) | 59 (64.13%) | 0.760[B] |
| Previous CABG, n (%) | 64 (13.31%) | 113 (20.55%) | **0.002**[B] | 18 (17.31%) | 47 (26.37%) | 0.080[B] | 18 (19.57%) | 22 (23.91%) | 0.475[B] |
| **CAD risk factors** | | | | | | | | | |
| Diabetes mellitus, n (%) | 174 (36.17%) | 222 (40.36%) | 0.168[B] | 44 (42.31%) | 107 (58.79%) | **0.007**[B] | 44 (47.83%) | 44 (47.83%) | 1.000[B] |
| - Insulin requiring, n (%) | 41 (8.52%) | 76 (13.82%) | **0.008**[B] | 17 (16.35%) | 49 (26.92%) | **0.041**[B] | 17 (18.48%) | 20 (21.74%) | 0.581[B] |
| Hypertension, n (%) | 422 (87.73%) | 495 (90.00%) | 0.247[B] | 88 (84.62%) | 170 (93.41%) | **0.016**[B] | 84 (91.30%) | 83 (90.22%) | 0.799[B] |
| Hyperlipidemia, n (%) | 411 (85.45%) | 460 (84.55%) | 0.686[B] | 92 (88.46%) | 153 (84.07%) | 0.308[B] | 82 (89.13%) | 78 (84.78%) | 0.381[B] |
| **Clinical presentation** | | | | | | | | | |
| - Stable angina | 230 (47.82%) | 277 (50.36%) | 0.415[B] | 41 (39.42%) | 72 (39.56%) | 0.982[B] | 36 (39.13%) | 37 (40.22%) | 0.880[B] |
| - Unstable angina | 148 (30.77%) | 172 (31.27%) | 0.862[B] | 27 (25.96%) | 56 (31.32%) | 0.339[B] | 25 (27.17%) | 29 (31.52%) | 0.517[B] |
| - NSTEMI | 86 (17.88%) | 96 (17.45%) | 0.828[B] | 32 (30.77%) | 48 (26.37%) | 0.426[B] | 27 (29.35%) | 24 (26.09%) | 0.621[B] |
| - STEMI | 17 (3.53%) | 5 (0.91%) | **0.004**[B] | 4 (3.85%) | 5 (2.75%) | 0.728[C] | 4 (4.35%) | 2 (2.17%) | 0.682[C] |
| Atrial fibrillation, n (%) | 49 (10.19%) | 93 (16.91%) | **0.002**[B] | 19 (18.27%) | 49 (26.92%) | 0.098[B] | 17 (18.48%) | 18 (19.57%) | 0.851[B] |
| Current smoker, n (%) | 98 (20.37%) | 124 (22.55%) | 0.397[B] | 17 (16.35%) | 25 (13.74%) | 0.549[B] | 13 (14.13%) | 14 (15.22%) | 0.835[B] |
| Family history of CAD, n (%) | 188 (39.17%) | 135 (25.67%) | **<0.001**[B] | 41 (39.42%) | 38 (22.62%) | **0.003**[B] | 39 (42.39%) | 17 (19.77%) | **0.001**[B] |
| **Concomitant disease and left ventricular ejection fraction** | | | | | | | | | |
| Pulmonary disease, n (%) | 40 (8.32%) | 52 (9.45%) | 0.522[B] | 8 (7.69%) | 18 (9.89%) | 0.534[B] | 8.00 (8.70%) | 7.00 (7.61%) | 0.788[B] |
| Peripheral artery disease, n (%) | 52 (10.81%) | 113 (20.55%) | **<0.001**[B] | 21 (20.19%) | 45 (24.73%) | 0.381[B] | 21.00 (22.83%) | 20.00 (21.74%) | 0.859[B] |
| LVEF%, Me (IQR) | 50.00 (44.00, 55.00) | 50.00 (45.00, 58.25) | **<0.001**[A] | 50 (39.50, 53.00) | 45.00 (37.00, 55.00) | 0.670[A] | 50.00 (39.50, 50.50) | 48.00 (38.00, 55.50) | 0.846[A] |

Abbreviations: BMI – body mass index; CABG – coronary artery bypass graft; CAD – coronary artery disease; CKD – chronic kidney disease; DAPT – dual antiplatelet therapy; DCB – drug-coated balloon; DES – drug-eluting stent; ISR – in-stent restenosis; IQR – interquartile range; LVEF – left ventricular ejection; Me – median; MI – myocardial infraction; MVD – multi-vessel disease; N/A – not applicable; n – number; NSTEMI – Non-ST-elevation myocardial infarction; STEMI – ST-elevation myocardial infarction.

*Note:* [A] – Wilcoxon rank sum test; [B] – Pearson's chi-squared test. [C] – Fisher's exact test.

### Angiographic and procedural data

Considering the frequency of thrombus followed by thrombectomy among patients with preserved kidney function, DES use was more frequent than DCB, though there were no such differences in the CKD group before and after PSM (Table 2). Stent implantation in the non-CKD group was performed in significantly tighter stenoses, and this relationship remained in the CKD group but disappeared after PSM (Table 2). Also, the left main artery was treated with a DES more often than a DCB in the non-CKD group, which was not reflected in the CKD group before and after PSM (Table 2). The original stent was longer, with a smaller diameter, in patients treated with a DCB in the non-CKD group, as well as in the CKD group before PSM, but was not statistically significant after PSM (Table 2). Focal restenosis and predilatation were observed more often among DCB-treated patients than those receiving a DES in all three groups (Table 2). The mean DES length used for PCI was significantly longer in the non-CKD group, whereas no statistical differences were found in the CKD group before and after PSM (Table 2). Residual restenosis was observed more often in the DCB than the DES group in non-CKD patients and in CKD patients after PSM (Table 2). The length of DAPT was significantly shorter in patients treated with a DCB rather than a DES in the non-CKD and CKD groups before and after PSM (Table 2).

### Long-term outcomes

There were no differences in the primary endpoint for TLR frequency during follow-up between patients treated with DES and DCB in the non-CKD group before (p = 0.873) and after PSM (p = 0.159) (Table 3 and Fig 2). Also, there were no differences in clinical outcomes between patients treated with DES and DCB for secondary endpoints in the CKD group before and after PSM for TVR (unmatched, p = 0.542; post-PSM, p = 0.134) (S2 Fig), MI (unmatched, p = 0.348; post-PSM, p = 0.183; S3 Fig), TV-MI (unmatched, p = 0.527; post-PSM, p = 0.528; S4 Fig), DOCE (unmatched, p = 0.325; post-PSM, p = 0.062; Fig 3), and cardiac death (unmatched, p = 0.855; post-PSM, p = 0.615; Fig 4) (Table 3). After excluding dialysis patients, there were no differences between patients treated with DES or DCB before (p = 0.348) and after PSM (p = 183) (S5 Fig). Also, when considering patients undergoing dialysotherapy, the differences remained irrelevant (p = 0.527 for the DES group and p = 0.528 for the DCB group) (S6 Fig).

### Thin strut drug-eluting stents vs. drug-coated balloons

Results of the Cox regression analysis evaluating the impact of thin-DES on TLR during follow-up in patients with preserved kidney function demonstrated that body-mass index > 30 kg/m² (HR = 1.98) and diabetes mellitus (HR = 1.82) were significantly related to poorer outcomes. Statistical significance was not confirmed for other assessed predictors in the CKD group before and after PSM (Table 4).

## Discussion

The study confirmed significant differences in general characteristics among CKD patients treated with PCI due to ISR, with DCB-treated patients being younger and presenting a greater burden of concomitant diseases and more frequent LVEF impairment before PSM. Moreover, ISR was initially observed in longer and wider stents and focal restenosis was more prevalent in CKD patients treated with a DCB than a thin-DES before PSM. After PSM, most differences lost their statistical significance. Also, differences were observed in the duration of DAPT, which resulted directly from treatment guidelines. Before and after PSM in the CKD group, there were no disparities in the post-procedure follow-up for primary and secondary endpoints between PCI treatment with DCB and thin-DES. Although there was a tendency for more frequent DOCE occurrence after PSM in the DCB group, this relationship did not reach statistical significance. Furthermore, the results showed no variations in long-term outcomes for DCB and thin-DES when the CKD group was divided into dialysis and non-dialysis patients.



Table 2. Angiographic, procedural, and medication data according to device type before and after propensity score matching.

| | Non-matched cohort, n=1,317 | | | | | | Propensity score-matched groups, n=184 | | |
| | Preserved kidney function, n=1,031 | | | CKD, n=286 | | | CKD, n=184 | | |
| | Thin-DES n=481 (46.65%) | DCB n=550 (53.35%) | p-value | Thin-DES n=104 (36.36%) | DCB n=182 (63.74%) | p-value | Thin-DES n=92 (50.00%) | DCB n=92 (50.00%) | p-value |
|---|---|---|---|---|---|---|---|---|---|
| **Angiography** | | | | | | | | | |
| One-vessel disease, n (%) | 276 (57.38%) | 306 (55.64%) | 0.573B | 64 (61.54%) | 86 (47.25%) | **0.020**B | 55 (59.78) | 42 (45.65%) | 0.055B |
| Two-vessel disease, n (%) | 147 (30.56%) | 159 (28.91%) | 0.562B | 24 (23.08%) | 60 (32.97%) | 0.077B | 21 (22.83%) | 30 (32.61%) | 0.138B |
| MVD, n (%) | 55 (11.43%) | 82 (14.91%) | 0.101B | 16 (15.38%) | 36 (19.78%) | 0.354B | 16 (22.83%) | 20 (21.74%) | 0.457B |
| Bifurcation, n (%) | 83 (17.26%) | 90 (16.36%) | 0.702B | 18 (17.31%) | 43 (23.63%) | 0.210B | 16 (17.39%) | 16 (17.39%) | 1.000B |
| Thrombus, n (%) | 14 (2.91%) | 4 (0.73%) | **0.008**C | 3 (2.88%) | 3 (1.65%) | 0.671C | 3 (3.26%) | 1 (1.09%) | 0.621B |
| Thrombectomy, n (%) | 5 (1.04%) | 0 (0%) | **0.022**C | 1 (0.96%) | 0 (0%) | 0.365C | 1 (1.09%) | 0 (0%) | 1.000B |
| Calcification, n (%) | 17 (3.53%) | 21 (3.82%) | 0.809B | 5 (4.81%) | 10 (5.49%) | 0.802B | 4 (4.35%) | 7 (7.61%) | 0.351B |
| Stenosis, %, Me (IQR) | 80.00 (80.90, 90.00) | 80.00 (70.00, 90.00) | **0.011**A | 85.00 (80.00, 90.00) | 80.00 (75.00, 90.00) | **0.028**A | 85.00 (80.00, 90.00) | 90.00 (80.00, 90.00) | 0.786A |
| **Target lesion** | | | | | | | | | |
| Left main, n (%) | 45 (9.36%) | 33 (6.00%) | **0.042**B | 10 (10.58%) | 20 (10.99%) | 0.914B | 11 (11.96%) | 8 (8.70%) | 0.467B |
| Left anterior descending, n (%) | 198 (41.16%) | 226 (41.09%) | 0.981B | 44 (42.31%) | 81 (44.51%) | 0.719B | 36 (39.13%) | 47 (51.09%) | 0.103B |
| Left circumflex, n (%) | 77 (16.01%) | 147 (26.73%) | **<0.001**B | 15 (14.42%) | 49 (26.92%) | **0.015**B | 13 (14.13%) | 23 (25%) | 0.063B |
| Right coronary artery, n (%) | 162 (33.68%) | 175 (31.82%) | 0.525B | 36 (34.62%) | 54 (29.67%) | 0.386B | 34 (36.96%) | 26 (28.26%) | 0.208B |
| Original stent-length, mm, Me (IQR) | 20.00 (16.00, 24.00) n=256 | 23.00 (18.00, 30.00) n=268 | **<0.001**A | 20.00 (15.00, 25.00) n=56 | 24.00 (18.00, 32.00) n=128 | **0.005**A | 20.00 (15.00, 28.00) n=46 | 23.50 (18.00, 32.00) n=68 | 0.083A |
| Original stent diameter, mm, Me (IQR) | 3.00 (3.00, 3.50) n=253 | 3.00 (2.75, 3.50) n=255 | **0.006**A | 3.13 (3.00, 3.50) n=56 | 3.00 (2.50, 3.50) n=127 | **0.048**A | 3.00 (3.00, 3.50) n=46 | 3.00 (2.50, 3.50) n=68 | 0.065 |
| **Type of ISR** | | | | | | | | | |
| Focal, n (%) | 205 (42.62%) | 299 (54.36%) | **<0.001**B | 42 (40.38%) | 102 (56.04%) | **0.011**B | 37 (40.22%) | 51 (55.43%) | **0.039**B |
| Diffuse, n (%) | 147 (30.56%) | 189 (34.36%) | 0.194B | 44 (42.31%) | 57 (31.32%) | 0.061B | 40 (43.48%) | 28 (30.43%) | 0.067B |
| Proliferative, n (%) | 123 (25.57%) | 45 (8.18%) | **<0.001**B | 18 (17.31%) | 16 (8.79%) | **0.032**B | 15 (16.30%) | 10 (10.87%) | 0.282B |
| Occlusive, n (%) | 6 (1.25%) | 17 (3.09%) | **0.046**B | 0 (0%) | 7 (3.85%) | 0.051C | 0 (0%) | 3 (3.26%) | 0.246C |
| **Balloon pre-dilatation** | | | | | | | | | |
| Predilatation, n (%) | 287 (62.94%) n=456 | 358 (88.18%) n=406 | **<0.001**B | 77 (74.04%) | 138 (91.39%) | **<0.001**B | 68 (73.91%) n=92 | 70 (87.74%) n=78 | **0.009**B |
| Length, mm, Me (IQR) | 15.00 (12.00, 20.00) n=276 | 15.00 (15.00, 20.00) n=378 | 0.154A | 15.00 (12.00, 20.00) n=58 | 15.00 (15.00, 20.00) n=144 | 0.760A | 15.00 (12.00, 20.00) n=48 | 15.00 (15.00, 20.00) n=76 | 0.670A |
| Diameter, mm, Me (IQR) | 3.00 (2.50, 3.50) n=287 | 3.00 (2.50, 3.50) n=392 | 0.906A | 3.00 (2.50, 3.50) n=60 | 3.00 (2.50, 3.50) n=146 | 0.830A | 3 (2.50, 3.50) n=51 | 3 (2.50, 3.50) n=76 | 0.726A |

*(Continued)*

**Table 2.** (Continued)

| | Non-matched cohort, n=1,317 | | | | | | Propensity score-matched groups, n=184 | | |
| --- | --- | --- | --- | --- | --- | --- | --- | --- | --- |
| | Preserved kidney function, n=1,031 | | | CKD, n=286 | | | CKD, n=184 | | |
| | Thin-DES n=481 (46.65%) | DCB n=550 (53.35%) | p-value | Thin-DES n=104 (36.36%) | DCB n=182 (63.74%) | p-value | Thin-DES n=92 (50.00%) | DCB n=92 (50.00%) | p-value |
| **Device data** | | | | | | | | | |
| Length, mm, Me (IQR) | 20.00 (15.00, 28.00) n=482 | 20.00 (15.00, 25.00) n=485 | 0.868[A] | 18.00 (15.00, 28.25) n=104 | 20.00 (16.50, 24.00) n=78 | 0.938[A] | 18.00 (15.00, 30.00) n=92 | 20.00 (17.00, 25.00) n=79 | 0.810[A] |
| Diameter, mm, Me (IQR) | 3.00 (3.00, 3.50) n=482 | 3.00 (2.50, 3.50) n=498 | **0.030[A]** | 3.00 (2.75, 3.50) n=104 | 3.00 (2.50, 3.50) n=78 | 0.605[A] | 3.00 (2.75, 3.50) n=92 | 3.00 (2.00, 3.50) n=81 | 0.093[A] |
| **Post-procedure** | | | | | | | | | |
| Residual stenosis, n (%) | 17 (3.53%) | 53 (9.64%) | **<0.001[B]** | 6 (5.77%) | 18 (9.89%) | 0.277[B] | 5 (5.43%) | 13 (14.13%) | **0.047** |
| TIMI-3, n (%) | 478 (99.38%) | 547 (99.45%) | 1.000[B] | 102 (98.08%) | 178 (97.80%) | 1.000[B] | 90 (97.83%) | 89 (96.74%) | 1.000 |
| Perforation, n (%) | 0 (0%) | 1 (0.18%) | 1.000[C] | 0 (0%) | 0 (0%) | 1.000[C] | 0 | 0 | – |
| Dissection, n (%) | 18 (3.74%) | 9 (1.64%) | **0.035[C]** | 3 (2.88%) | 8 (4.40%) | 0.751[C] | 3 (3.26%) | 6 (6.52%) | 0.497 |
| No Reflow, n (%) | 4 (0.83%) | 3 (0.55%) | 0.711[C] | 0 (0%) | 2 (1.10%) | 0.536[C] | 0 | 0 | – |
| **Intracoronary imaging and drug therapy** | | | | | | | | | |
| Use of intracoronary imaging, n (%) | 22 (4.57%) | 27 (4.91%) | 0.801[B] | 3 (2.88%) | 11 (6.04%) | 0.234[C] | 3 (3.26%) | 5 (5.43%) | 0.720[C] |
| Glycoprotein IIb/IIIa inhibitors, n (%) | 13 (2.70%) | 2 (0.36%) | **0.002[C]** | 2 (1.92%) | 4 (2.2%) | 1.000[C] | 2 (2.17%) | 2 (2.17%) | 1.000[C] |
| Length of DAPT, months, Me (IQR) | 12.00 (12.00, 12.00) n=479 | 12.00 (6.00, 12.00) n=543 | **<0.001[A]** | 12.00 (12.00, 12.00) n=103 | 12.00 (6.00, 12.00) n=180 | **<0.001[A]** | 12.00 (12.00, 12.00) n=91 | 12.00 (6.00, 12.00) n=92 | **0.001[A]** |

Abbreviations: CKD – chronic kidney disease; DAPT – dual antiplatelet therapy; DCB – drug-coated balloon; DES – drug-eluting stent; ISR – in-stent restenosis; IQR – interquartile range; Me – median; MVD – multi-vessel disease; N/A – not applicable; n, number; TIMI – Thrombolysis in Myocardial Infarction risk score

Note: [A] – Wilcoxon rank sum test; [B] – Pearson's chi-squared test. [C] – Fisher's exact test.

**Table 3. Survival analysis for each device before and after propensity score matching.**

| | Crude analysis (n = 286) | | | | Propensity score analysis (n = 184) | | | |
|---|---|---|---|---|---|---|---|---|
| Chronic kidney disease | | | | | | | | |
| | Thin-DES n = 104 | DCB n = 182 | HR (95% CI)[1] | p-value | Thin-DES n = 92 | DCB n = 92 | HR (95% CI)[1] | p-value |
| **TLR, n (%)** | 11 (10.58%) | 18 (9.89%) | 0.94 (0.44–2.00) | 0.873 | 8 (8.70%) | 12 (13.04%) | 0.52 (0.21–1.29) | 0.159 |
| **TVR, n (%)** | 16 (15.38%) | 26 (14.29%) | 0.82 (0.44–1.55) | 0.542 | 13 (14.13%) | 14 (15.22%) | 0.54 (0.24–1.01) | 0.134 |
| **MI, n (%)** | 11 (10.58%) | 23 (12.64%) | 0.71 (0.34–1.46) | 0.348 | 10 (10.87%) | 13 (14.13%) | 0.56 (0.24–1.32) | 0.183 |
| **TV-MI, n (%)** | 4 (3.85%) | 8 (4.40%) | 0.67 (0.19–2.31) | 0.527 | 3 (3.26%) | 3 (3.26%) | 0.56 (0.09–3.39) | 0.528 |
| **CV death, n (%)** | 3 (2.88%) | 6 (3.3%) | 0.88 (0.22–3.55) | 0.855 | 2 (2.17%) | 3 (3.26%) | 0.63 (0.10–3.81) | 0.615 |
| **DOCE, n (%)** | 13 (12.50%) | 26 (14.29%) | 0.71 (0.36–1.40) | 0.325 | 10 (10.87%) | 15 (16.30%) | 0.45 (0.19–1.04) | 0.062 |

Abbreviations: CI – confidence interval; CV – cardiovascular; DCB – drug-coated balloon; DES – drug-eluting stent; DOCE – device-oriented composite endpoint; HR – hazard ratio; n – number; MI – myocardial infarction; TLR – target lesion revascularization; TV – target vessel; TVR – target vessel revascularization.

1 Thin-DES treated was the exposure, and DCB was the reference.

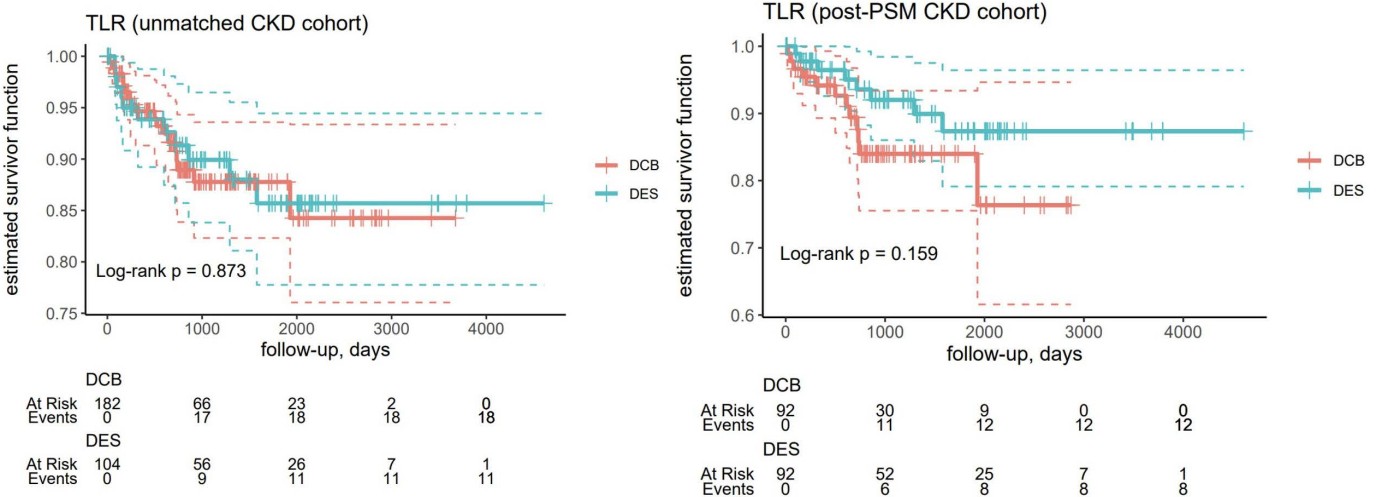

**Fig 2. Kaplan-Meyer curves for the estimated survival function of target vessel revascularization (TLR) in the unmatched chronic kidney disease (CKD) (n = 286, left) and post-PSM CKD cohorts (n = 184, right).** The dashed lines represent the 95% confidence intervals, while the horizontal marks indicate the patient censoring events (here and below).

In the overall population of patients treated for DES-ISR, no differences were demonstrated between thin-DES and DCB use, with some results merely showing a trend toward worse outcomes with DCB [8,12,14]. It can be assumed that in a specific population, such as those with renal failure, variations in results and long-term outcomes can be expected between those treated percutaneously with DCB and thin-strut DES due to ISR. This thesis results from the number of different mechanisms of atherosclerosis and restenosis in this group of patients. CKD patients represent a specific cohort in which several potential mechanisms of vascular injury lead to accelerated progression of atherosclerosis. Among such mechanisms are mineral bone derangements expressing with intensified vascular calcifications, chemical modification of lipoproteins, such as glycation, oxidation, and carbamylation, coexisting with very-low-density lipoprotein accumulation,

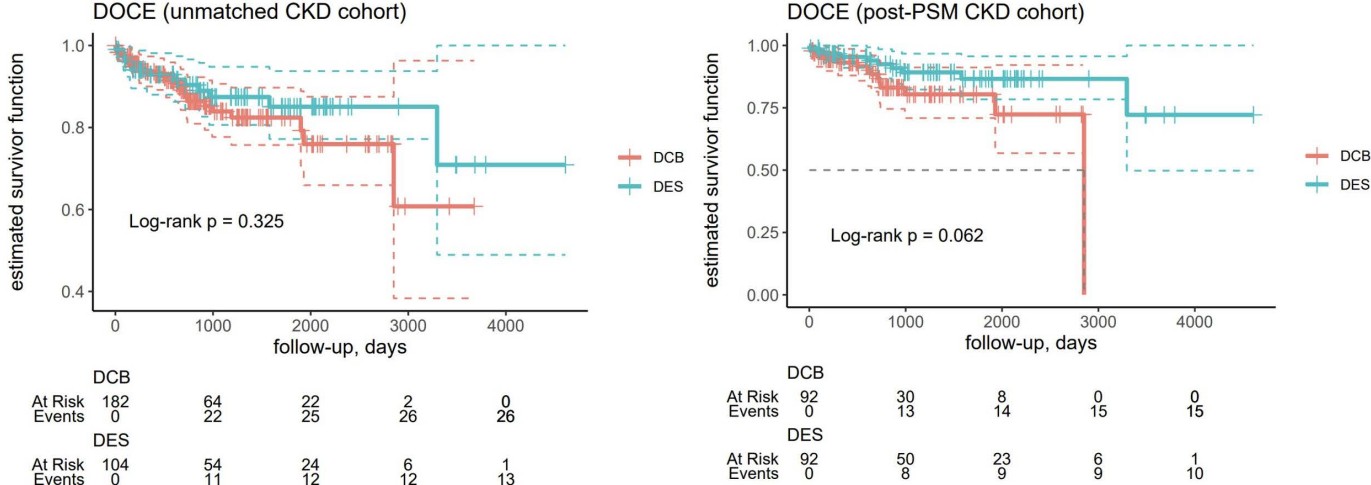

**Fig 3. Kaplan-Meyer curves for the estimated survival function of the device-oriented composite endpoint (DOCE) in the unmatched chronic kidney disease (CKD) (n = 286, left) and post-PSM CKD cohorts (n = 184, right).**

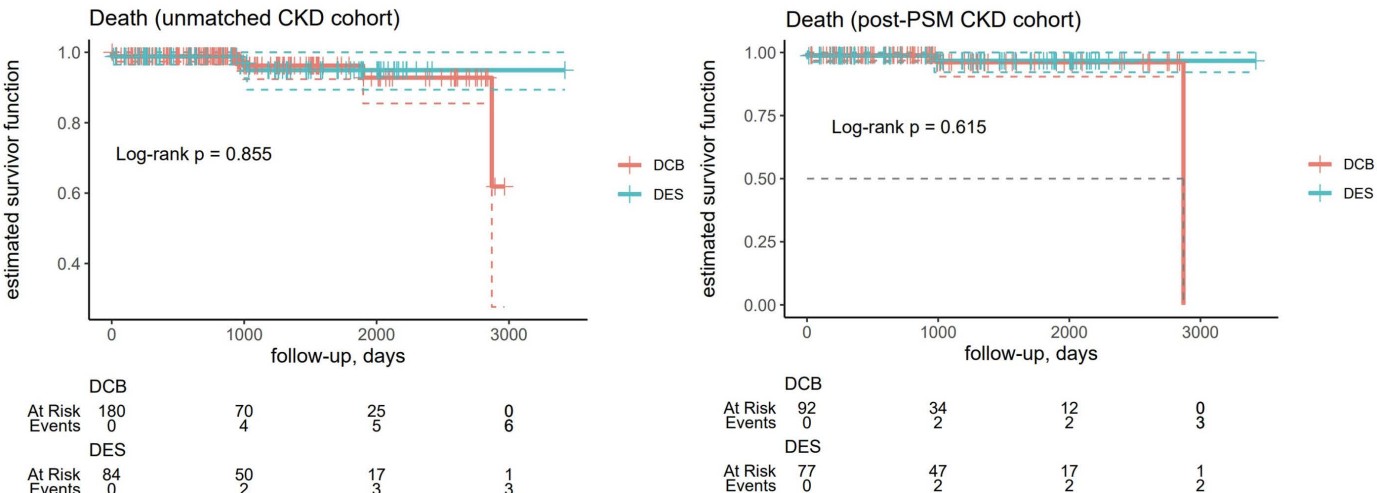

**Fig 4. Kaplan-Meyer curves for the estimated survival function of cardiovascular death in the unmatched chronic kidney disease (CKD) (n = 286, left) and post-PSM CKD cohorts (n = 184, right).**

and loss of the protective function of high-density lipoproteins. Additional mechanisms include increased levels of circulating C-reactive protein and cytokines, an activated phenotype of circulating monocytes and resident vascular cells, increased synthesis of inflammation-triggered reactive oxygen species, and impairment of endothelial function caused by increased levels of inorganic phosphates and albuminuria, which was recently found to be a stronger risk factor for mortality than eGFR [32].

While the current findings do not demonstrate statistically significant differences between thin-strut drug-eluting stents and drug-coated balloons, they underscore a state of clinical equipoise in this high-risk population with chronic kidney disease and in-stent restenosis. This equipoise provides a strong rationale for future adequately powered randomized controlled trials to definitively evaluate comparative efficacy and safety.

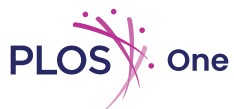

Table 4. Results of Cox regression analysis evaluating the impact of thin-strut drug-eluting stents on target lesion revascularization during follow-up in patients with chronic kidney disease and preserved kidney function, stratified by subgroups.

| Subgroup | Non-matched cohort, n = 1,317 | | | | | | Propensity score-matched group | | |
|---|---|---|---|---|---|---|---|---|---|
| | Preserved kidney function, n = 1,031 | | | CKD, n = 286 | | | CKD, n = 184 | | |
| | HR² | 95% CI | P-Value | HR² | 95% CI | P-Value | HR² | 95% CI | P-Value |
| Male | 1.11 | 0.73–1.68 | 0.612 | 1.09 | 0.43–2.78 | 0.855 | 0.65 | 0.22–1.87 | 0.422 |
| Female | 1.01 | 0.54–1.88 | 0.982 | 0.76 | 0.22–2.71 | 0.680 | 0.37 | 0.07–2.03 | 0.254 |
| Age ≥ 65 years | 1.26 | 0.77–2.06 | 0.366 | 0.94 | 0.41–2.12 | 0.874 | 0.57 | 0.21–1.53 | 0.263 |
| Age < 65 years | 0.90 | 0.56–1.45 | 0.672 | 0.76 | 0.08–6.81 | 0.807 | 0.40 | 0.04–3.91 | 0.433 |
| BMI ≥ 30 | 1.98 | 1.04–3.79 | **0.038** | 2.78 | 0.51–15.25 | 0.239 | 2.57 | 0.27–24.77 | 0.414 |
| BMI < 30 | 0.69 | 0.42–1.13 | 0.144 | 0.46 | 0.15–1.45 | 0.186 | 0.20 | 0.04–1.03 | 0.054 |
| DM | 1.82 | 1.04–3.19 | **0.035** | 1.14 | 0.33–3.91 | 0.834 | 0.78 | 0.19–3.11 | 0.720 |
| No DM | 0.79 | 0.51–1.22 | 0.283 | 0.72 | 0.28–1.87 | 0.497 | 0.40 | 0.12–1.34 | 0.135 |
| Previous MI | 0.96 | 0.63–1.47 | 0.859 | 0.86 | 0.29–2.51 | 0.777 | 0.46 | 0.13–1.60 | 0.224 |
| No previous MI | 1.36 | 0.75–2.45 | 0.298 | 0.98 | 0.34–2.85 | 0.973 | 0.59 | 0.16–2.22 | 0.436 |
| Bifurcation lesion | 0.73 | 0.33–1.61 | 0.431 | 2.06 | 0.51–8.25 | 0.308 | 1.16 | 0.19–7.01 | 0.870 |
| No Bifurcation lesion | 1.19 | 0.81–1.74 | 0.374 | 0.70 | 0.28–1.75 | 0.447 | 0.39 | 0.13–1.15 | 0.087 |
| Left main PCI | 0.65 | 0.21–2.01 | 0.449 | 1.54 | 0.22–10.99 | 0.667 | 0.68 | 0.04–10.98 | 0.785 |
| No left main PCI | 1.13 | 0.79–1.63 | 0.496 | 0.87 | 0.38–1.98 | 0.747 | 0.46 | 0.17–1.26 | 0.130 |
| Diameter ALL ≥ 3 mm | 0.91 | 0.60–1.37 | 0.642 | 0.94 | 0.38–2.31 | 0.888 | 0.49 | 0.161–1.52 | 0.219 |
| Diameter ALL < 3 mm | 1.61 | 0.82–3.16 | 0.167 | 0.86 | 0.21–3.60 | 0.836 | 0.48 | 0.10–2.41 | 0.376 |
| Length ALL ≥ 22 mm | 1.02 | 0.55–1.90 | 0.952 | 0.52 | 0.12–2.20 | 0.378 | 0.37 | 0.08–1.66 | 0.195 |
| Length ALL < 22 mm | 1.05 | 0.68–1.61 | 0.831 | 1.28 | 0.51–3.21 | 0.594 | 0.67 | 0.20–2.23 | 0.517 |
| Dialysis | Not applicable (no dialysis) | | | 1.55 | 0.10–24.93 | 0.757 | Not applicable (only one in the DES group had TLR) | | |
| No dialysis | 1.08 | 0.77-1.53 | 0.654 | 0.89 | 0.40–1.94 | 0.760 | 0.46 | 0.18–1.18 | 0.106 |

Abbreviations: BMI – body mass index; CI – confidence interval; CKD – chronic kidney disease; DES – drug-eluting stent; DM – diabetes mellitus; LM – left main; HR – hazard ratio; MI – myocardial infarction; PCI – percutaneous coronary intervention TLR – target lesion revascularization.

1 Thin – DES vs. DCB.

Previously published studies have confirmed a higher incidence of recurrent ISR and MACE in hemodialysis (HD) patients compared to non-HD patients treated with DCB due to ISR [33]. In that study, TLR incidence during up to three years of follow-up reached 41% in the HD group vs. 9.6% in the non-HD (p < 0.0001) [33]. Similar long-term observations with a follow-up period of up to 1800 days were presented in a study published by Lee et al., which showed a significantly higher incidence of target vessel failure (TVF), repeated revascularization, and all-cause mortality in patients treated with DCB due to ISR and impaired renal function compared to those with preserved renal function [15].

At the beginning of this century, attempts were made to reduce the frequency of restenosis in patients with impaired renal function using various devices, one of them being intracoronary radiation, although they did not show any improvement and were even characterized by a number of periprocedural complications and ultimately performed worse than conventional treatment [34]. Studies published so far have confirmed that DES use in dialysis patients yields a significant decrease in the risk of mortality, MACE, and TL/VR [35]. However, several studies did not confirm these findings and showed no additive effect of DES treatment over bare-metal stents [36]. The introduction of new stents with thin layers did not significantly improve the treatment outcomes of CKD patients in the general population [37]. Similarly, no effect of the type of antimitotic drug released was demonstrated on the clinical outcomes of PCI treatment in patients with CKD in the general population [38]. The data available in the literature regarding the influence of the presence and severity of chronic

renal failure on ISR treatment using thin-strut DES and DCB are very limited. Therefore, the present analysis is pioneering and indicates that renal failure has an equal impact on the long-term results of treatment using both techniques. Confirmation of this thesis requires observations on larger groups of patients from randomized studies.

## Study limitations

This investigation is subject to several inherent limitations that must be acknowledged when interpreting the results, particularly with respect to the evaluation of long-term clinical outcomes following percutaneous coronary intervention for in-stent restenosis in patients with chronic kidney disease. Chief among these is the constrained sample size of the propensity score-matched cohort, comprising 184 patients (92 per treatment arm), coupled with low cumulative event rates for the assessed endpoints (e.g., 13.6% for the device-oriented composite endpoint [DOCE] and 3.3% for target vessel myocardial infarction). This configuration yields substantial underpowering, as demonstrated by post-hoc power analyses utilizing the Schoenfeld approximation for the Cox proportional hazards model. For instance, with 25 observed DOCE events and a hazard ratio (HR) of 0.45, the study achieved only 51% power to detect this effect at $\alpha = 0.05$ (two-sided), conferring a high risk of Type II error – specifically, a greater than 49% probability of failing to identify a true difference despite the observed trend (p = 0.062). Comparable post-hoc powers for secondary endpoints were even lower, ranging from 7% for cardiovascular death to 36% for target vessel revascularization, further amplifying the potential for false-negative conclusions.

An a priori power calculation, absent from the original study design, would have been indispensable for ensuring adequate sensitivity. Hypothetically, for planning future studies in similar populations, detecting an HR of 0.45 (aligned with the observed point estimate) with 80% power and $\alpha = 0.05$, under an assumed event rate of 13.6%, would necessitate approximately 49 events and a total sample of 362 patients (181 per group). For a more conservative anticipated effect size (HR = 0.50), the requirements escalate to 65 events and 481 patients (241 per group). These estimates highlight the current study's insufficiency in event accrual and cohort size, which may obscure clinically relevant benefits of thin-strut drug-eluting stents over drug-coated balloons, even amid numerical trends favoring the former.

Beyond statistical power, the retrospective nature of the registry introduces potential selection and confounding biases, notwithstanding propensity score matching for key baseline covariates such as age, diabetes mellitus, and lesion morphology. Residual confounding from unmeasured factors, including the underlying mechanisms of restenosis (due to the lack of routine intravascular imaging) and operator-dependent treatment selections, cannot be fully excluded. Furthermore, the absence of centralized core laboratory adjudication for angiographic parameters may compromise the precision of lesion characterizations. The findings' generalizability is confined to patients with chronic kidney disease and in-stent restenosis, potentially limiting applicability to broader populations. Although the median follow-up duration of 959 days is clinically pertinent, it may not capture late-emerging disparities in outcomes.

Furthermore, as an exploratory, hypothesis-generating study evaluating multiple endpoints without corrections for multiplicity, the results are susceptible to inflated Type I error rates, potentially leading to spurious associations. Larger, prospective randomized trials with prespecified adjustments for multiple testing are warranted to validate these preliminary observations.

## Conclusions

In CKD patients undergoing PCI for ISR, thin-DES treatment was associated with a numerical reduction in TLR, TVR, and DOCE compared with DCB. However, these differences did not achieve statistical significance in the crude or PSM analyses.

## Supporting information

**S1 Fig. Covariate balance before and after propensity score matching.**
(JPG)



**S2 Fig. Kaplan -Meyer curves for estimated survival function of TVR in unmatched CKD cohort (N = 286, left) and post-PSM CKD cohort (N = 184, right).**
(JPG)

**S3 Fig. Kaplan -Meyer curves for estimated survival function of MI in unmatched CKD cohort (N = 286, left) and post-PSM CKD cohort (N = 184, right).**
(JPG)

**S4 Fig. Kaplan -Meyer curves for estimated survival function of MI-TVR in unmatched CKD cohort (N = 286, left) and post-PSM CKD cohort (N = 184, right).**
(JPG)

**S5 Fig. Kaplan -Meyer curves for estimated survival function of TLR in unmatched CKD cohort (N = 263, left) and post-PSM CKD cohort (N = 169, right) for non-dialyzed patients.** The dashed lines represent the 95% confidence intervals, while the horizontal marks indicate the patient censoring events.
(JPG)

**S6 Fig. Kaplan -Meyer curves for estimated survival function of TLR in unmatched CKD cohort (N = 23, left) and post-PSM CKD cohort (N = 15, right) for dialyzed patients.** The dashed lines represent the 95% confidence intervals, while the horizontal marks indicate the patient censoring events.
(JPG)

## Acknowledgments

The Dragon Registry was initiated on the Club 30 Scientific Platform of the Polish Cardiac Society.

## Author contributions

**Conceptualization:** Rafał Januszek, Sylwia Iwańczyk, Fabrizio D'Ascenzo, Łukasz Kuźma, Brunon Tomasiewicz, Piotr Niezgoda, Robert J. Gil, Bernardo Cortese.

**Data curation:** Rafał Januszek, Marta Chamera, Sylwia Iwańczyk, Fabrizio D'Ascenzo, Łukasz Kuźma, Brunon Tomasiewicz, Piotr Niezgoda, Rafał Wolny, Mariusz Kowalewski, Maciej Wybraniec, Krzysztof Reczuch, Sławomir Dobrzycki, Ovidio De Filippo, Artur Pawlik, Karol Kasprzycki, Tomasz Pawłowski, Kamil Skowron, Stanisław Bartuś, Maciej Lesiak, Mariusz Gąsior, Jacek Kubica, Piotr Waciński, Francesco Bruno, Wojciech Wojakowski, Wojciech Wańha.

**Formal analysis:** Rafał Januszek, Mariusz Kowalewski.

**Funding acquisition:** Rafał Wolny.

**Investigation:** Rafał Januszek, Sylwia Iwańczyk, Fabrizio D'Ascenzo, Łukasz Kuźma, Piotr Niezgoda, Rafał Wolny, Maciej Wybraniec, Krzysztof Reczuch, Sławomir Dobrzycki, Ovidio De Filippo, Artur Pawlik, Karol Kasprzycki, Tomasz Pawłowski, Kamil Skowron, Maciej Lesiak, Mariusz Gąsior, Jacek Kubica, Robert J. Gil, Piotr Waciński, Francesco Bruno, Bernardo Cortese, Wojciech Wojakowski, Wojciech Wańha.

**Methodology:** Rafał Januszek, Marta Chamera, Sylwia Iwańczyk, Fabrizio D'Ascenzo, Brunon Tomasiewicz, Piotr Niezgoda, Rafał Wolny, Mariusz Kowalewski, Maciej Wybraniec, Stanisław Bartuś.

**Software:** Rafał Januszek, Maciej Wybraniec.

**Supervision:** Rafał Januszek, Piotr Niezgoda, Rafał Wolny, Krzysztof Reczuch, Ovidio De Filippo, Robert J. Gil, Bernardo Cortese.



**Validation:** Rafał Januszek, Fabrizio D'Ascenzo, Łukasz Kuźma, Brunon Tomasiewicz, Piotr Niezgoda, Rafał Wolny, Mariusz Kowalewski, Maciej Wybraniec, Krzysztof Reczuch, Ovidio De Filippo, Robert J. Gil, Bernardo Cortese, Wojciech Wojakowski.

**Visualization:** Piotr Niezgoda, Rafał Wolny, Mariusz Kowalewski.

**Writing – original draft:** Rafał Januszek, Marta Chamera, Sylwia Iwańczyk, Fabrizio D'Ascenzo, Łukasz Kuźma, Brunon Tomasiewicz, Piotr Niezgoda, Rafał Wolny, Mariusz Kowalewski, Maciej Wybraniec, Artur Pawlik, Karol Kasprzycki, Tomasz Pawłowski, Kamil Skowron, Stanisław Bartuś, Maciej Lesiak, Mariusz Gąsior, Jacek Kubica, Piotr Waciński, Francesco Bruno, Wojciech Wojakowski, Wojciech Wańha.

**Writing – review & editing:** Rafał Januszek, Sylwia Iwańczyk, Brunon Tomasiewicz, Rafał Wolny, Maciej Wybraniec, Sławomir Dobrzycki, Ovidio De Filippo, Robert J. Gil, Bernardo Cortese.

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
