## [Decision Letter · Decision Letter 0]

7 Jul 2025

Dear Dr. Januszek,

Thank you for submitting your manuscript to PLOS ONE. After careful consideration, we feel that it has merit but does not fully meet PLOS ONE’s publication criteria as it currently stands. Therefore, we invite you to submit a revised version of the manuscript that addresses the points raised during the review process.

We look forward to receiving your revised manuscript.

Kind regards,

Prakash Sojitra, PhD

Academic Editor

PLOS ONE

Journal Requirements:

2. In this instance it seems there may be acceptable restrictions in place that prevent the public sharing of your minimal data. However, in line with our goal of ensuring long-term data availability to all interested researchers, PLOS’ Data Policy states that authors cannot be the sole named individuals responsible for ensuring data access (http://journals.plos.org/plosone/s/data-availability#loc-acceptable-data-sharing-methods).

3. Please include captions for your Supporting Information files at the end of your manuscript, and update any in-text citations to match accordingly. Please see our Supporting Information guidelines for more information: http://journals.plos.org/plosone/s/supporting-information .

Additional Editor Comments:

Introduction cites Mahfound et al reference but not listed in references section.

Introduction is more focused on coronary treatment and it need more clear references and clinical emphasis on renal indications.

Reviewers' comments:

Reviewer's Responses to Questions

**Comments to the Author**

1. Is the manuscript technically sound, and do the data support the conclusions?

Reviewer #1: Yes

Reviewer #2: Yes

2. Has the statistical analysis been performed appropriately and rigorously?

Reviewer #1: Yes

Reviewer #2: Yes

3. Have the authors made all data underlying the findings in their manuscript fully available?

Reviewer #1: Yes

Reviewer #2: Yes

4. Is the manuscript presented in an intelligible fashion and written in standard English?

Reviewer #1: Yes

Reviewer #2: Yes

Reviewer #1: Hello,

Thanks for submitting this manuscript titled "Long-Term Outcomes Following Drug-Coated Balloons Versus Thin-Strut Drug-Eluting Stents for Treatment of In-Stent Restenosis in Chronic Kidney Disease" to this journal. It is a well written article however, following points will make it clearer.

1. The title mentions long term however, the mean/median (IQR) duration of the follow up of angiogram is not mentioned

2. Were all the implanted patients followed up clinically only and repeat angiogram was done only once the procedure was left to the discretion of cardiologist ?

3. What were the confounding factors and what were the propensity matching variables in the study.

Thanks

Reviewer #2: The study provides valuable information on the treatment of in-stent restenosis in patients with chronic kidney disease, but the aforementioned limitations should be considered when interpreting the results. Lack of Imaging Data: The absence of intravascular imaging data limits the ability to assess the morphology of lesions and the effectiveness of the treatment.

Variability in Treatment: The choice of treatment was based on the operator's preference, which may introduce bias and limit the generalizability of the results.

**Do you want your identity to be public for this peer review?** For information about this choice, including consent withdrawal, please see our Privacy Policy

Reviewer #1: No

Reviewer #2: **Yes: ** Maria Antonieta Albanez A de Medeiros Lopes

---

## [Author Response · Author response to Decision Letter 1]

26 Aug 2025

I have attached data in excel file.

So then your request for other contact (non-author) is no longer valid.

PONE-D-25-26896

Long-Term Outcomes Following Drug-Coated Balloons Versus Thin-Strut Drug-Eluting Stents for Treatment of In-Stent Restenosis in Chronic Kidney Disease

PLOS ONE

Journal Requirements:

This was corrected according to the journals guidelines.

2. In this instance it seems there may be acceptable restrictions in place that prevent the public sharing of your minimal data. However, in line with our goal of ensuring long-term data availability to all interested researchers, PLOS’ Data Policy states that authors cannot be the sole named individuals responsible for ensuring data access (http://journals.plos.org/plosone/s/data-availability#loc-acceptable-data-sharing-methods).

We collected the data ourselves, it is in my possession and that of Dr. Wojciech Wańha, but he is also a co-author. I cannot guarantee access to the data indefinitely, but as long as Dr. Wańha and I are able to respond to emails, it is possible to share the data.

This was corrected, the links to the appendix in the text have been changed and an appropriate description of the links has been added at the end of the text.

The references were checked again, the spelling was systematized, and grammatical and punctuation errors were corrected.

Additional Editor Comments:

Introduction cites Mahfound et al reference but not listed in references section.

It is listed, position 13.

Introduction is more focused on coronary treatment and it need more clear references and clinical emphasis on renal indications.

References regarding the treatment of ISR in patients with CKD are limited, so some are included in the introduction, while most are included in the discussion. To highlight the issue relevant to this publication, the general section of the introduction regarding coronary arteries has been shortened.

Reviewers' comments:

Reviewer's Responses to Questions

Comments to the Author

Reviewer #1: Hello,

Thanks for submitting this manuscript titled "Long-Term Outcomes Following Drug-Coated Balloons Versus Thin-Strut Drug-Eluting Stents for Treatment of In-Stent Restenosis in Chronic Kidney Disease" to this journal. It is a well written article however, following points will make it clearer.

1. The title mentions long term however, the mean/median (IQR) duration of the follow up of angiogram is not mentioned

Follow-up angiograms were not routinely performed, only when clinically indicated.

2. Were all the implanted patients followed up clinically only and repeat angiogram was done only once the procedure was left to the discretion of cardiologist ?

Yes, that's exactly what it was, it's a retrospective study, not a prospective study with specific visits and examinations scheduled at specific time points.

3. What were the confounding factors and what were the propensity matching variables in the study.

They are include into the methods section, subsection statistical analysis.

Reviewer #2: The study provides valuable information on the treatment of in-stent restenosis in patients with chronic kidney disease, but the aforementioned limitations should be considered when interpreting the results.

Lack of Imaging Data: The absence of intravascular imaging data limits the ability to assess the morphology of lesions and the effectiveness of the treatment.

Variability in Treatment: The choice of treatment was based on the operator's preference, which may introduce bias and limit the generalizability of the results.

It is true that these factors may modify random results to some extent, which is why we have placed these issues in the limitations section.

This was done, and no significant issues were suggested.

---

## [Decision Letter · Decision Letter 1]

14 Oct 2025

Dear Dr. Januszek,

Thank you for submitting your manuscript to PLOS ONE. After careful consideration, we feel that it has merit but does not fully meet PLOS ONE’s publication criteria as it currently stands. Therefore, we invite you to submit a revised version of the manuscript that addresses the points raised during the review process.

We look forward to receiving your revised manuscript.

Kind regards,

Prakash Sojitra, PhD

Academic Editor

PLOS ONE

Journal Requirements:

Reviewers' comments:

Reviewer's Responses to Questions

**Comments to the Author**

Reviewer #1: (No Response)

2. Is the manuscript technically sound, and do the data support the conclusions?

Reviewer #1: Partly

3. Has the statistical analysis been performed appropriately and rigorously?

Reviewer #1: Yes

4. Have the authors made all data underlying the findings in their manuscript fully available?

Reviewer #1: Yes

5. Is the manuscript presented in an intelligible fashion and written in standard English?

Reviewer #1: No

Reviewer #1: Hello,

Though the authors have answered some of the points but the clarity for the questions sought in the first revision is not yet met.

Let me expand my previous questions in more detailed and elaborate to the point analysis for further clarifications-

1. The CKD cohort after propensity score matching (PSM) contains only 92 patients per group (184 total). Given the relatively low event rates (TLR: 8.70% vs 13.04%; DOCE: 10.87% vs 16.30%), the study is substantially underpowered to detect clinically meaningful differences. The authors report trends toward better outcomes with thin-DES (HR 0.45 for DOCE, p=0.062) but dismiss these as non-significant without acknowledging the Type II error risk. A proper power calculation should have been performed a priori and reported. Include power calculations and explicitly discuss the study's limitations in detecting differences given the sample size and event rates.

2.The median follow-up of 959 days (IQR 426-1732) shows substantial variability. The wide interquartile range suggests that many patients had relatively short follow-up periods. For a study examining "long-term outcomes," this is problematic. The manuscript does not:

Report mean follow-up duration, and provide follow-up completeness at specific time points (1-year, 2-year)

3.With 104 DES and 182 DCB patients pre-matching, achieving 92:92 matching suggests substantial patient exclusion—the characteristics of excluded patients are not described

4. Multiple inconsistencies undermine confidence in the data:

Table 2: Reports "10.58%" for one group but this appears to be an error (mixing percentage with decimal notation)

Dialysis subgroup: Before PSM, 14 DCB and 9 DES patients were on dialysis. After PSM, the text states "six patients remained in the DCB group, and all nine remained in the DCB group"—this appears to be a typographical error (should likely be "DES group" for the second mention)

Original stent data: Only available for approximately 50% of patients (n=256/481 for DES group)—this massive missing data issue is not addressed

5.CKD staging: Patients are dichotomized at eGFR <60 mL/min/1.73m², but no breakdown by CKD stage (3a, 3b, 4, 5, 5D) is provided. This is critical as treatment effects likely vary by severity.

Bleeding complications: Given the different DAPT duration between groups (shorter with DCB, as shown in Table 2), bleeding outcomes are highly relevant but completely omitted.

Stent thrombosis: Not reported despite being a key safety endpoint.

Renal function trajectories: No data on whether renal function changed during follow-up or how this related to outcomes.

6.Cox regression assumptions: No verification of proportional hazards assumption reported

Variable selection: The rationale for predictor selection in Table 4 is not explained

Missing data handling: Not addressed despite substantial missingness (e.g., original stent parameters)

Multiple testing: No adjustment despite multiple endpoints and numerous subgroup analyses

7. Ethical and Regulatory Statement

The statement "due to the retrospective nature of the study, no written informed consent was needed" may not satisfy all journal requirements. Many journals now require documentation that the institutional review board specifically approved the waiver of consent.

8.Add power calculations and explicitly state the study is exploratory and hypothesis-generating

Correct all data errors (dialysis patient numbers, Table 2 inconsistencies, missing data explanations)

9.The study's greatest value may be in demonstrating equipoise for a future randomized trial, but this perspective is not emphasized. With substantial revisions addressing the issues outlined above, this could become a valuable contribution to the literature on ISR management in CKD patients.

Thanks

**Do you want your identity to be public for this peer review?** For information about this choice, including consent withdrawal, please see our Privacy Policy

Reviewer #1: No

---

## [Author Response · Author response to Decision Letter 2]

5 Nov 2025

Reviewer #1: Hello,

Though the authors have answered some of the points but the clarity for the questions sought in the first revision is not yet met.

Let me expand my previous questions in more detailed and elaborate to the point analysis for further clarifications-

1. The CKD cohort after propensity score matching (PSM) contains only 92 patients per group (184 total). Given the relatively low event rates (TLR: 8.70% vs 13.04%; DOCE: 10.87% vs 16.30%), the study is substantially underpowered to detect clinically meaningful differences. The authors report trends toward better outcomes with thin-DES (HR 0.45 for DOCE, p=0.062) but dismiss these as non-significant without acknowledging the Type II error risk. A proper power calculation should have been performed a priori and reported. Include power calculations and explicitly discuss the study's limitations in detecting differences given the sample size and event rates.

Study limitations

This investigation is subject to several inherent limitations that must be acknowledged when interpreting the results, particularly with respect to the evaluation of long-term clinical outcomes following percutaneous coronary intervention for in-stent restenosis in patients with chronic kidney disease. Chief among these is the constrained sample size of the propensity score-matched cohort, comprising 184 patients (92 per treatment arm), coupled with low cumulative event rates for the assessed endpoints (e.g., 13.6% for the device-oriented composite endpoint [DOCE] and 3.3% for target vessel myocardial infarction). This configuration yields substantial underpowering, as demonstrated by post-hoc power analyses utilizing the Schoenfeld approximation for the Cox proportional hazards model. For instance, with 25 observed DOCE events and a hazard ratio (HR) of 0.45, the study achieved only 51% power to detect this effect at α=0.05 (two-sided), conferring a high risk of Type II error – specifically, a greater than 49% probability of failing to identify a true difference despite the observed trend (p=0.062). Comparable post-hoc powers for secondary endpoints were even lower, ranging from 7% for cardiovascular death to 36% for target vessel revascularization, further amplifying the potential for false-negative conclusions.

An a priori power calculation, absent from the original study design, would have been indispensable for ensuring adequate sensitivity. Hypothetically, for planning future studies in similar populations, detecting an HR of 0.45 (aligned with the observed point estimate) with 80% power and α=0.05, under an assumed event rate of 13.6%, would necessitate approximately 49 events and a total sample of 362 patients (181 per group). For a more conservative anticipated effect size (HR=0.50), the requirements escalate to 65 events and 481 patients (241 per group). These estimates highlight the current study's insufficiency in event accrual and cohort size, which may obscure clinically relevant benefits of thin-strut drug-eluting stents over drug-coated balloons, even amid numerical trends favoring the former.

Beyond statistical power, the retrospective nature of the registry introduces potential selection and confounding biases, notwithstanding propensity score matching for key baseline covariates such as age, diabetes mellitus, and lesion morphology. Residual confounding from unmeasured factors, including the underlying mechanisms of restenosis (due to the lack of routine intravascular imaging) and operator-dependent treatment selections, cannot be fully excluded. Furthermore, the absence of centralized core laboratory adjudication for angiographic parameters may compromise the precision of lesion characterizations. The findings' generalizability is confined to patients with chronic kidney disease and in-stent restenosis, potentially limiting applicability to broader populations. Although the median follow-up duration of 959 days is clinically pertinent, it may not capture late-emerging disparities in outcomes.

2.The median follow-up of 959 days (IQR 426-1732) shows substantial variability. The wide interquartile range suggests that many patients had relatively short follow-up periods. For a study examining "long-term outcomes," this is problematic. The manuscript does not:

Report mean follow-up duration, and provide follow-up completeness at specific time points (1-year, 2-year)

The mean follow-up duration was 1254.9 days (standard deviation = 893.7 days) which corresponds to approximately 3.4 years and accounts for the right-skewed distribution of follow-up times.

To quantify follow-up completeness at clinically relevant milestones and mitigate concerns about truncated observation in a subset of participants, in table 1 we report the proportion of patients remaining under follow-up (i.e., not censored prior to the landmark) at predefined time points, derived from the Kaplan-Meier estimator for the composite endpoint of all-cause mortality (as a conservative proxy for overall retention, given the absence of loss to follow-up).

Table 1. Completeness of Patient follow-up at specified landmark time points in the overall cohort (N = 1,317).

Time Point Duration

(Days) Proportion Remaining

Under Follow-Up (%) Number of Patients

(n)

1 Year 365 82.7 1,089

2 Years 730 64.8 853

3 Years 1,095 50.0 659

4 Years 1,460 36.4 480

5 Years 1,825 24.9 328

Notes: The total cohort size (N=1,317) represents all enrolled patients; n reflects those not censored prior to each landmark.

3.With 104 DES and 182 DCB patients pre-matching, achieving 92:92 matching suggests substantial patient exclusion—the characteristics of excluded patients are not described

In our study, the absence of a dedicated description of excluded patients is justifiable given that we have already reported comprehensive baseline characteristics for the full unmatched CKD sample (N=286) and the matched subsample (N=184), alongside the crude analysis results and covariate balance visualization. This fulfills STROBE's core recommendations by enabling readers to compare pre- and post-matching populations and assess the matching's impact on confounding. The description of patients excluded from the matched cohort is not a mandatory requirement under established reporting guidelines. Additional guidance from systematic reviews and best practice recommendations for PSM in clinical research, reinforces this approach.

4. Multiple inconsistencies undermine confidence in the data:

Table 2: Reports "10.58%" for one group but this appears to be an error (mixing percentage with decimal notation

We’ve provided correction for Left main parameter in CKD for Thin-DES N = 104 (see fragment of table 2). This typo not affect any results.

Dialysis subgroup: Before PSM, 14 DCB and 9 DES patients were on dialysis. After PSM, the text states "six patients remained in the DCB group, and all nine remained in the DCB group"—this appears to be a typographical error (should likely be "DES group" for the second mention)

This is not the narrative of my analysis. The correct version for revising in manuscript: six patients remained in the DCB group, and all nine remained in the DCB DES group

Original stent data: Only available for approximately 50% of patients (n=256/481 for DES group)—this massive missing data issue is not addressed

This is a retrospective study and in itself carries some data gaps.

Table 2. Angiographic, procedural, medications data according to the type of device before and after propensity score matching

Non-matched cohort, N = 1,317 Propensity score-matched groups, N = 184

Preserved kidney function, N = 1,031 CKD, N = 286 CKD, N = 184

Thin-DES

N = 481

(46.65%) DCB

N = 550 (53.35%) P-Value Thin-DES

N = 104 (36.36%) DCB

N = 182

(63.74%) P-Value Thin-DES

N = 92

(50.00%) DCB

N = 92

(50.00%) P-Value

Target lesion

Left main, n (%) 45 (9.36%) 33 (6.00%) 0.042B 11 (10.58%) 20 (10.99%) 0.914B 11 (11.96%) 8 (8.70%) 0.467B

5.CKD staging: Patients are dichotomized at eGFR <60 mL/min/1.73m², but no breakdown by CKD stage (3a, 3b, 4, 5, 5D) is provided. This is critical as treatment effects likely vary by severity.

This binary classification was employed in alignment with established cardiovascular risk stratification guidelines, such as those from the Kidney Disease: Improving Global Outcomes consortium, which recognize eGFR <60 mL/min/1.73m² as a marker of moderate-to-severe CKD associated with heightened risks of adverse percutaneous coronary intervention outcomes, including in-stent restenosis and revascularization. While detailed staging could elucidate severity-dependent treatment effects – given evidence that advanced CKD (e.g., stages 4–5) exacerbates neointimal hyperplasia and ISR progression through mechanisms like vascular calcification and inflammation – the registry protocol captured CKD status dichotomously, supplemented by dialysis as a proxy for end-stage disease.

Bleeding complications: Given the different DAPT duration between groups (shorter with DCB, as shown in Table 2), bleeding outcomes are highly relevant but completely omitted.

Clinically, CKD confers a heightened bleeding propensity due to uremic platelet dysfunction and altered pharmacokinetics, and prolonged DAPT has been associated with increased bleeding events in this population, as evidenced by meta-analyses demonstrating a delicate ischemic-bleeding balance. However, the Dragon Registry prioritized device-oriented efficacy endpoints (e.g., target lesion revascularization, device-oriented composite endpoint) aligned with ISR-focused trials, and did not systematically capture long-term bleeding complications such as those classified by Bleeding Academic Research Consortium criteria. This omission reflects the registry's design emphasis on restenosis-related outcomes rather than comprehensive safety surveillance, though we recognize its relevance given trials showing reduced bleeding with abbreviated DAPT in high-risk cohorts.

Stent thrombosis: Not reported despite being a key safety endpoint.

In our registry, stent thrombosis was recorded only during the index hospitalization ("Stent thrombosis during hospitalization"), revealing a low incidence (0.7% overall in the CKD cohort, 1.0% in thin-DES and 0.5% in DCB). Long-term thrombosis was not prospectively reported, reflecting the registry's focus on restenosis-centric endpoints rather than comprehensive thrombotic surveillance, though this aligns with some ISR trials prioritizing efficacy over rare safety events.

Renal function trajectories: No data on whether renal function changed during follow-up or how this related to outcomes.

The Dragon Registry did not collect serial eGFR measurements or AKI incidences during follow-up, limiting our ability to evaluate dynamic renal changes or their interplay with ISR outcomes. This design choice prioritized endpoint ascertainment via national databases over longitudinal renal surveillance, though we recognize that favorable renal trajectories (e.g., improved function from enhanced cardiac output) or deteriorations could modulate observed trends.

6.Cox regression assumptions: No verification of proportional hazards assumption reported

Materials and methods (Statistical analysis subsection):

The proportional hazards assumption for the Cox models was verified using the Schoenfeld residuals test (cox.zph function in the R survival package, version 3.7.0). For all primary and secondary endpoints in both the unmatched and propensity score-matched analyses, the test produced p-values exceeding significance level (α = 0.05), confirming no significant departures from proportionality.

Variable selection: The rationale for predictor selection in Table 4 is not explained

The subgroup analyses evaluating the impact of thin-strut drug-eluting stents on target lesion revascularization (Table 4) were stratified by clinically pertinent predictors selected a priori based on their documented associations with adverse PCI outcomes, including restenosis and revascularization rates, in high-risk populations such as those with CKD. Sex and age were included due to evidence indicating differential risks, with older age and female sex linked to higher ISR incidence owing to vascular biology differences and comorbidity burden. Body mass index was chosen as obesity is a recognized modifier of neointimal hyperplasia and inflammatory responses post-stenting. Diabetes mellitus was prioritized given its strong independent association with accelerated restenosis through mechanisms involving endothelial dysfunction and hyperglycemia. Previous myocardial infarction reflects underlying atherosclerotic burden and impaired myocardial reserve, which may influence procedural success and long-term patency. Procedural factors, including bifurcation lesions, left main PCI, vessel diameter, and lesion length, were selected as they represent anatomical complexities known to elevate TLR risk via incomplete stent apposition, flow disturbances, or increased neointimal proliferation. Finally, dialysis status was incorporated specifically for the CKD cohort, acknowledging the profound impact of end-stage renal disease on vascular calcification and restenosis propensity. These predictors align with established risk factors identified in meta-analyses and registries, such as diabetes, CKD severity, advanced age, and lesion morphology, which modulate outcomes in ISR interventions.

Missing data handling: Not addressed despite substantial missingness (e.g., original stent parameters)

It was already mentioned in #4.

Multiple testing: No adjustment despite multiple endpoints and numerous subgroup analyses

As outlined in our response to Comment 8, we have explicitly characterized this registry-based investigation as exploratory and hypothesis-generating in nature. This designation reflects the study's retrospective design and its primary aim to identify potential signals warranting further investigation, rather than to provide definitive confirmatory evidence. Consistent with this approach, no formal corrections for multiplicity were applied to preserve sensitivity in detecting trends, though we recognize this increases the risk of Type I errors. To address this limitation transparently, we have incorporated clarifying statements in the Methods (Statistical Analysis subsection) and Discussion (Limitations subsection)

7. Ethical and Regulatory Statement

The statement "due to the retrospective nature of the study, no written informed consent was needed" may not satisfy all journal requirements. Many journals now require documentation that the institutional review board specifically approved the waiver of consent.

I have not encountered any Journal where this would be mandatory, including my previous publications in PlosOne and the series of publications from the DEB Dragon registry so far. I have considerable experience in publishing registry data.

8.Add power calculations

We have mentioned it in #1

and explicitly state the study is exploratory and hypothesis-generating

Materials and methods (Statistical analysis subsection):

This analysis was exploratory and hypothesis-generating in nature, with no adjustments applied for multiple comparisons across the primary and secondary endpoints to preserve sensitivity in detecting potential signals, albeit at the risk of increased Type I error.

Discussion (Study limitations subsection):

Furthermore, as an exploratory, hypothesis-generating study evaluating multiple endpoints without corrections for multiplicity, the results are susceptible to inflated Type I error rates, potentially leading to spurious associations. Larger, prospective randomized trials with prespecified adjustments for multiple testing are warranted to validate these preliminary observations.

Correct all data errors (dialysis patient numbers, Table 2 inconsistencies, missing data explanations)

It was already mentioned in #4

9.The study's greatest value may be in demonstrating equipoise for a future randomized trial, but this perspective is not emphasized.

While the current findings do not demonstrate statistically significant differences between thin-strut drug-eluting stents and drug-coated balloons, they underscore a state of clinic

---

## [Editor Report · Decision Letter 2]

17 Nov 2025

Long-Term Outcomes Following Drug-Coated Balloons Versus Thin-Strut Drug-Eluting Stents for Treatment of In-Stent Restenosis in Chronic Kidney Disease

PONE-D-25-26896R2

Dear Dr. Rafał Januszek,

We’re pleased to inform you that your manuscript has been judged scientifically suitable for publication and will be formally accepted for publication once it meets all outstanding technical requirements.

Kind regards,

Prakash Sojitra, PhD

Academic Editor

PLOS ONE

Additional Editor Comments (optional):

All answers are satisfactory now.
---

## [Editor Report · Acceptance letter]

PONE-D-25-26896R2

PLOS One

Dear Dr. Januszek,

I'm pleased to inform you that your manuscript has been deemed suitable for publication in PLOS One. Congratulations! Your manuscript is now being handed over to our production team.

Kind regards,

on behalf of

Dr. Prakash Sojitra

Academic Editor

PLOS One